# Using Community-Based Participatory Research Methods to Inform the Development of Medically Tailored Food Kits for Hispanic/*Latine* Adults with Hypertension: A Qualitative Study

**DOI:** 10.3390/nu15163600

**Published:** 2023-08-17

**Authors:** Ambria Crusan, Kerrie Roozen, Clara Godoy-Henderson, Kathy Zamarripa, Anayeli Remache

**Affiliations:** 1Department of Nutrition and Dietetics, Henrietta Schmoll School of Health Sciences, St. Catherine University, St. Paul, MN 55105, USA; 2Department of Health Services Research, Policy and Administration, School of Public Health, University of Minnesota, Minneapolis, MN 55455, USA; 3Department of Biology, School of Arts, Humanities, and Sciences, St. Catherine University, St. Paul, MN 55105, USA; 4Department of Psychology, School of Arts, Humanities, and Sciences, St. Catherine University, St. Paul, MN 55105, USA

**Keywords:** Hispanic, *Latine*, medically tailored meal kit, hypertension, fruits, vegetables, DASH eating plan, cardiovascular disease, culture

## Abstract

The Dietary Approaches to Stop Hypertension (DASH) eating plan is the most effective dietary intervention for cardiovascular disease (CVD), but it excludes the consideration of culture and cost. The Hispanic/*Latine* population is disproportionately affected by CVD, with risks increasing if persons are accustomed to a Westernized diet. This research aims to understand the cultural dietary practices aligned with a DASH eating plan and the social determinants of health impacting fruit and vegetable (F/V) consumption among immigrant Hispanic/*Latine* individuals at a community-based clinic in Minnesota. Utilizing community-based participatory research methods, a community survey informed the development of DASH-focused, medically tailored food kits of varying F/V modalities. Qualitative feedback was sought out regarding the kits when presented to 15 individuals during in-depth interview sessions to validate the cultural appropriateness of food kits for clinical use. Box A was the highest rated kit (66.7%) and consisted of fresh F/V. The average F/V consumption per day was 2.6 ± 1.4 servings. The food insecurity questionnaires showed high/marginal (40%), low (53.3%), and very low (6.7%) food security. The barriers to consuming F/V were money, time, and transportation. Understanding cultural dietary practices related to the DASH eating plan is necessary to mitigate CVD risk and provide inclusive medical nutrition therapy for Hispanic/*Latine* populations.

## 1. Introduction

Hispanic and *Latine* populations are at a disproportionately higher risk for poor health outcomes, such as cardiovascular disease (CVD), due to inconsistent food access, documentation status, and limited access to healthcare and social support [1,2,3]. Key risk factors for CVD include high blood pressure, dyslipidemia, diabetes mellitus, obesity, smoking, consumption of a high-fat diet, and physical inactivity [4]. Hispanic/*Latine* individuals face rates of overweight or obesity at a higher rate than the general population [5], increasing the risk for morbidity and mortality [6]. Moreover, immigrants who adopt Westernized diets through acculturation are at an even greater risk of developing chronic diseases, particularly hypertension (HTN) and diabetes [1]. A phenomenon known as the Healthy Immigrant Effect theorizes that people who have migrated to the United States (U.S.) tend to have better physical health than domestic-born individuals. Yet, this advantage decreases as the immigrant’s duration of residence in the U.S. increases [7]. Links between the length of residence in the U.S. and CVD risk for Hispanic/*Latine* individuals were also found, as women who migrated to the U.S. within 10 years had a lower CVD risk, whereas men exhibited the lowest CVD risk after living in the U.S. for more than 10 years [1].

The literature indicates that the Dietary Approaches to Stop Hypertension (DASH) eating plan is a nutritional intervention for reducing systolic blood pressure by up to 11 mmHg for individuals with HTN [4]; however, this eating pattern has been predominantly researched in association with a Westernized diet, and its effectiveness has not been assessed for a wide variety of cultural food patterns [8,9]. The DASH eating plan encourages the consumption of 4–5 servings of both fruits and vegetables and recommends dietary staples such as whole grains, vegetable oils, legumes, fish, and poultry [10] for a clinically meaningful lowering of blood pressure [11]. Fruits and vegetables (F/V) are emphasized by the DASH eating plan, as they are a rich dietary source of carotenoids and other antioxidants that work to mitigate oxidative stress [12].

The strategies outlined for evidence-based guidance to manage HTN do not consider social determinants of health, including low socioeconomic status or barriers to access and the availability of nutrient-dense foods such as fresh F/V. Instead, the DASH eating plan favors dietary patterns of white, affluent individuals [8]. For example, making dietary changes to consume an abundance and variety of F/V, fatty fish, and nuts/seeds can be a barrier for low-income individuals due to the combination of cost, availability, and access [8,13]. Individuals may also lack culturally relevant recipes when F/V access or familiarity is low [11]. A narrative review by Livingstone et al. noted the potential for inequities in diet quality and health outcomes to increase for ethnic minorities because nutritional interventions do not use community-based participatory practices that listen to community voices to develop the intervention, leaving their needs unmet [14].

A recent expansion in the use of medically tailored meal kits has shown to be effective in managing chronic conditions via diet [15,16,17,18]. However, there is still a need to develop culturally appropriate nutrition interventions for individuals managing chronic diseases with limited resources, providing equity in CVD treatment. A gap in the literature exists regarding the cultural acceptability of nutritional interventions such as medically tailored meal kits for Hispanic/*Latine* communities. The primary aim of this research is to use qualitative interviews to better understand F/V preferences and the existing barriers to obtaining culturally appropriate F/V that align with the DASH eating plan for immigrant Hispanic/*Latine* individuals with HTN. The secondary aim is to better understand participant demographics, food security status, and dietary F/V intake patterns. This study hypothesizes that barriers due to affordability, access, and availability of F/V will be prevalent for this low-income study population. Moreover, there will be a greater acceptability of food kits curated from high-preference F/V than those of moderate- to high-preference F/V previously identified in a community survey.

## 2. Materials and Methods

### 2.1. Study Design

Community-based participatory research methods were used to better understand which F/V to include in a medically tailored food kit for Hispanic/*Latine* individuals. This project included a conceptualization component, which was carried out via a survey informed by the literature, which was completed by 50 patients or providers self-identifying as Hispanic/*Latine* from a community health clinic to understand F/V preferences. This survey informed the formative intervention for 1-on-1 interviews to learn about the perceptions of the DASH-appropriate food kit design. Six F/V boxes consisting of culturally appropriate F/V that were indicated as higher preference were developed. High-preference items were denoted as F/V that over 75% of the survey population indicated as preferred in the survey. A separate staple items kit, including items such as rice, *masa* flour, beans, *tortillas*, onions, cilantro, and garlic, was added to support meal development and increased consumption of F/V, which aligns with the DASH eating plan recommendations for whole grains, legumes, and non-salt seasonings. The F/V were displayed in varying modalities (fresh, frozen, and canned), with three F/V boxes containing culturally appropriate, high-preference F/V, and three boxes with culturally appropriate, high-carotenoid F/V ranging from moderate to high preference (as shown in Table 1). The F/V boxes and staple kits were referenced in the interview to gain further insights on individual preferences.

Participants also completed demographic and medical questionnaires consisting of validated measures, such as questions derived from the PhenX toolkit [19], the PROMIS Scale (v1.2) Global Health instrument [20], the U.S. Department of Agriculture’s Six-Item Short Form of the Food Security Survey [21], and the 10-item Dietary Screener Questionnaire (DSQ) [22], prior to starting the semi-structured qualitative interview as a part of action research.

### 2.2. Participants and Recruitment

In June of 2022, a small population of Hispanic/*Latine* patients and providers (*n* = 50 participants) were recruited onsite at a health clinic in Minnesota to complete the community-based survey. The inclusion criteria for the survey were as follows: aged ≥18 years, a patient or provider (such as nurse, medical interpreter, or community health worker) at the local community health clinic, and self-identifying as Hispanic/*Latine*. For the 1 h interviews taking place between October 2022 and January 2023, participants were recruited through convenience sampling from a group who had previously participated in the survey and expressed interest in being contacted for a follow-up interview. To reach the intended target population, additional participants were recruited from a community health clinic through referrals from community health workers. Participants who expressed interest in learning more about the study were then referred to the study coordinator. Out of the 60 participants who were contacted, 23 were recruited to participate in the study. Of these participants, a total of 15 consented to the study and completed the interview; 8 participants canceled or did not show up for the scheduled interview. The interviews were conducted with 15 participants who were ≥18 years, a patient at the community health clinic, self-identifying as Hispanic/*Latine*, and managing a chronic disease, as indicated in their survey.

This study was approved by St. Catherine University’s Institutional Review Board #1744 and was conducted within the approved parameters. All participants provided written or verbal informed consent prior to engaging in the survey and/or the interview. Emphasis was placed on there being no connection between their right to receive healthcare and their participation in this study. Because the principal investigator provides healthcare services at the community health clinic, researchers ensured that no prior relationships were made with participants prior to starting the study.

### 2.3. Data Collection

Survey data from the conceptualization phase informed the questions that were developed for a semi-structured, ethnographic interview, which was utilized to ensure consistent content was explored with each participant. Ethnographic research was conducted to address a knowledge gap in individuals’ perspectives regarding culturally appropriate F/V for chronic disease management, allowing for the researchers to learn about preferences and barriers for a migrant Hispanic/*Latine* population. This method is particularly effective to provide formative interventions and cross-cultural understanding [23]. Each interview included an introduction to the purpose of the study, the researcher’s role in the community health clinic and university, the objectives of the interview being conducted, the anticipated duration of the interview, the right to refuse to answer any questions that cause discomfort, and empowering the participants to share their opinions with the intent of learning from the patients.

An exploration of the participants’ views on the contents of each box was conducted via interview questions examining participant preferences and perceptions of health. The research team also utilized probing and follow-up questions to ensure understanding between participants and the research team or to expand on examples provided during the interview.

The interviews were conducted virtually in Spanish by two to three female members of the research team (A.C., K.R., C.G.-H., and A.R.), two of which have expertise in the field of nutrition and dietetics (A.C. and K.R.), one with previous experience in qualitative interviewing strategies (A.C.), and two who served as interpreters (A.R. and C.G.-H.) when needed. The interviews were conducted in a quiet and private room at St. Catherine University. The Standards for Reporting Qualitative Research Checklist was used to ensure that the study methods were reported accurately [24].

### 2.4. Data Analysis

This intervention utilized 15 individual interviews to gather feedback and input on the acceptability of the contents in the food kits and to obtain perceptions of health. All of the interviews were audio recorded, transcribed verbatim into Microsoft Word version 16.71 (Microsoft, Redmond, WA, USA), and individually translated from Spanish to English, where appropriate. Data were deidentified upon transcription and participant identifiers were used during the validation of the interviews and the thematic analysis. Themes were assessed via the seven-step Framework Method, which is a systematic approach used in healthcare research that often suits research interprofessional teams with mixed levels of qualitative research experience [25]. This method of analysis is well suited to the thematic analysis of semi-structured interviews, allowing for the expression of opinions and the voicing of preferences.

In accordance with the stepwise outline of the Framework Method [26], the interviews were transcribed verbatim following the interviews that took place between October and January to complete step 1 after the conclusion of all 15 interviews. For interviews conducted in Spanish, the conversation was transcribed verbatim in Spanish, translated to English, and then translated back to Spanish to ensure accuracy. Step 2, or familiarization with the interview, occurred after each of the interviews were transcribed with the research team using both the audio and transcribed interviews for reference. Following the familiarization, over 3 weeks in January of 2023, 5 independent researchers from multiple disciplines (A.C., K.R., C.G-.H., K.Z., and A.R.) conducted step 3 by coding the transcripts in groups of 3–4 participants, using their individual interpretations to determine codes. After coding the first 4 transcripts in a 5-day timespan, the team used consensus building to develop thematic details for application, completing step 4, and then used step 5 to apply the framework in the indexing of the additional 11 interviews reviewed in the next 2 weeks through the working framework. Additional themes or subthemes were added, coding the other interviews, if necessary, and the first 4 interviews were reviewed to find data aligning with the theme. Subthemes were determined based on variable perspectives connected to each theme. This iterative process of familiarization, coding, and indexing allowed for flexibility within the coding, providing accuracy of participant perspectives. All data were then charted to the framework matrix (step 6), and the analysis team met in February 2023 to interpret the data outcomes (step 7) after all the data were charted to the matrix. Microsoft Excel version 16.71 (Microsoft, Redmond, WA, U.S.) was used to generate a spreadsheet with charted thematic outputs. Data interpretation was discussed as a team using research perceptions and theoretical concepts related to nutrition and public health research. Table 2 and Figure 1 and Figure 2 were created in Rstudio version 2023.03.0.

The findings of our study were verified for trustworthiness as one author (A.C.) trained the research team on the coding framework and independently verified each reviewer’s coding strategies to establish consistency across 5 independent coders (A.C., K.R., C.G.-H., K.Z., and A.R.). Moreover, the research team discussed each interview after coding, ensuring that proper procedures were followed while developing and indexing the themes throughout the analysis. Any discrepancies were challenged, and a consensus was reached by the team by reviewing the matrix.

### 2.5. Questionnaires

The exploratory analysis to better understand participant demographics, dietary F/V intake patterns, and social determinants of health was conducted via quantitative methods. Data from the PhenX toolkit [19] were used to obtain participants’ age, biological sex, gender identity, birthplace, race/ethnicity, educational attainment, employment status, and English proficiency. The measures included in the PhenX toolkit are validated for use across populations and have wide acceptability to reliably make comparisons between studies [19]. The Patient-Reported Outcomes Measurement Information System (PROMIS) Scale V 1.2, Global Health instrument—which can be used to collect information regarding physical, mental, and social health in adults living with chronic conditions—was used to understand participants’ health perceptions [20].

The U.S. Department of Agriculture’s (USDA) Six-Item Short Form of the Food Security Survey [21] was used to determine food security status, disaggregated via binary and tiered food security [27]. The USDA’s Six-Item Short Form of the Food Security Survey is an effective tool to identify households facing varying levels of food security with “reasonably high specificity and sensitivity and minimal bias” [21]. Using the binary measure of food security, food insecurity was determined using the reported analysis measures provided by the survey [21]. Additionally, the 10-item DSQ [22] was utilized to determine the number of daily servings of fruits and vegetables the study participants consumed in the past month. Scoring procedures were conducted via the Conversion of Frequency Response to Times per Day provided by the National Cancer Institute’s Data Processing & Scoring Procedures [28]. The DSQ was used as it is a validated F/V recall tool. Research indicates that its application is effective for use in low-income ethnic minority groups. Moreover, it does not require previous diet education for portion sizes, as the quantities are reported in cups and converted to portion sizes by the researchers [29].

Quantitative study data were collected and managed using the Research Electronic Data Capture (REDCap) electronic tool hosted at St. Catherine University [30,31,32]. REDCap is a secure web-based software platform designed to support data capture for research studies, providing (1) an intuitive interface for validated data capture; (2) audit trails for tracking data manipulation and export procedures; (3) automated export procedures for seamless data downloads to common statistical packages; and (4) procedures for data integration and interoperability with external sources [31]. REDCap was used to determine the means and standard deviations reported in this study.

## 3. Results

### 3.1. Participant Demographics

The mean age of the 12 women (80%) and 3 (20%) men recruited for this study was 54.8 ± 8.8, and 73.4% of the individuals were born in Mexico. The races of the participants were reported as black (*n* = 1), white (*n* = 12), and Native American Aztec (*n* = 2), with all of the participants being of Hispanic ethnicity (Mexican, Puerto Rican, Guatemalan, Ecuadorian, and Venezuelan). The average family size living with the participants in their household was 3.6 ± 2.1, and the reported time spent cooking per day ranged from 15 min to 2 h. The 10-item DSQ assessed F/V intake and showed a total average consumption of 2.6 ± 1.4 servings of F/V per day. The average for the reported fruit intake was much lower than that of vegetables (0.2 ± 0.5 versus 2.4 ± 1.3, respectively). The average daily servings of the total F/V according to demographics are shown in Table 2.

In assessing food security status using the binary measure (*n* = 14), we found that food insecurity was reported in 50% of the sample. For ranges of food security and food insecurity in our sample (*n* = 14), high or marginal food security was reported by 40%, low food security was reported by 53.3%, and very low food security was reported by 6.7%. One participant refused to report their food security status. Figure 1 represents food insecurity reporting, detailing question 2, which asks, “(I/we) couldn’t afford to eat balanced meals. Was that often, sometimes, or never true for (you/your household) in the last 12 months?” and can be reflective of the ability to obtain F/V. The average number of servings of F/V consumed per day was higher among individuals who reported food insecurity (3.3 ± 1.4) compared to those who did not report food security (1.9 ± 1.0). Moreover, we found a positive perceived health status, which was reflected in the PROMIS Scale V 1.2—Global Health scores [20]; the mean score for Global Mental Health was 48.7 ± 3.7, and for Global Physical Health, the mean score was 47.9 ± 4.5, with non-significant differences of 1.3 and 2.1 below the U.S. reference population (*p* = 0.99 and *p* = 0.98, respectively).

### 3.2. Box Preference and Cultural Appropriateness

Using the list of high-preference fruits and vegetables determined by the community question survey, the items in Boxes A, B, and C were developed and portioned to show approximately 50 servings of F/V. The participants ranked the boxes from highest preference (1) to lowest preference (6). Box A, with all fresh F/V, was the highest preference for 66.7% of the participants; Box B, containing a mix of fresh and frozen F/V, was the highest preference for 13.3% of the participants; and Box C, with fresh, frozen, and canned F/V, was not ranked as any of the participants’ highest preference. Box C was most commonly ranked as the fifth preference out of the six boxes. Box A, curated with all high-preference and fresh F/V, was considered culturally appropriate by all 15 participants. One participant stated, “That’s why I prefer [Box A]. They are products that I consume, and that I’m used to consuming, instead of the box with the canned items, I don’t use them”. Another participant expressed preferences for certain items in Box A, noting, “[Box A] that had the chayote (squash), the carrots and all of that. The spinach and everything, all of that we consume in Mexico”. The participants’ top box preferences are shown in Figure 2.

Boxes D, E, and F were developed using F/V from the moderate- to high-preference F/V options that are also high in carotenoids and then portioned to show approximately 50 servings of F/V. Box D, containing all fresh F/V, was the highest preference for 20% of the participants, whereas Box E, containing fresh and frozen F/V, and Box F, which contained fresh, frozen, and canned items, were not ranked as any of the participants’ highest preference. In fact, 46.7% of the participants ranked Box F as their lowest preference. Of the 15 participants, 8 deemed Box D as culturally appropriate. The participants who deemed Box D as culturally inappropriate reported that the inclusion of kale and butternut squash was not representative of the foods that are commonly consumed within their cultural cuisine.

Overall, the foods that were more frequently indicated as “missing” from the food kits that were appropriate for the participants’ culture were zucchini (*n* = 3), blueberries (*n* = 3), cabbage (*n* = 3), mushrooms (*n* = 2), corn (*n* = 2), and raspberries (*n* = 2). Other preferences voiced were green versus red apples, plantains versus bananas, and fresh pineapple over canned pineapple.

### 3.3. Themes Identified

The identification of the following four higher-order themes emerged in relation to the participants’ F/V preferences and challenges in obtaining their preferred F/V: a preference for fresh F/V over frozen or canned items; barriers to obtaining F/V were abundant and most commonly included time, money, and transportation; a challenge in utilizing the F/V if there were not any staple foods (rice, beans, and onions) to complement the use of the items; and individuals who showed variable degrees of health literacy. More specifically, the knowledge of cooking/preparing F/V was high; however, the portion sizes of F/V and the quantities recommended for good health were unclear. The thematic map including the identified subthemes and supporting quotes for each theme/subtheme are shown in Table 3.

#### 3.3.1. Theme 1: Preference for Fresh Fruits and Vegetables

There was a strong overall preference indicated for fresh F/V over frozen or canned items (83% said no canned or frozen items), which was reflected in the first preference being the highest for Boxes A and D, containing all fresh items, and had participant confirmation. Participant 7 noted the following:

“The best option is the most natural, because of the chemicals that the canned goods have. But in reality, all the boxes had something good, because tomatoes, apples, bananas, carrots, kale, spinach, broccoli, all of those are good things.”

A nuanced perspective of this quote details multiple items that are not typically sold frozen (apples, tomatoes, bananas, and kale) and voices concern for the quality of canned items. The term that the participants frequently associated with fresh F/V was “natural” (*n* = 5), with frozen foods being associated with preservation and canned items noted as containing increased “chemicals”, sodium, or sugar. Of note, corn and tomatoes were mentioned as appropriate canned items when seasonality limits access to fresh options.

Aligning with the theme of individuals showing a preference for fresh F/V based on personal food preferences and/or cultural upbringing, Participant 1 acknowledged the following:

“I know that a lot of people in the United States use a lot of canned foods. They use it because it’s more convenient and fast, and we don’t, we usually grow it and like it more natural [in Mexico].”

The majority of the participants noted that canned items were used as a last resort.

#### 3.3.2. Theme 2: Time, Money, and Transportation Are Barriers to Obtaining Fruits and Vegetables

Three barriers to obtaining F/V were identified as common for individuals in our sample, which include (1) the time taken to obtain F/V (*n* = 5), (2) money to purchase F/V (*n* = 9), and (3) transportation to access the grocery store (*n* = 4). Figure 3 details individual responses to the barriers indicated by the participants, also showing any overlap where more than one barrier may be present for an individual. Multiple participants mentioned challenges with time due to work schedules, necessary medical appointments for health management, and the need to support their family, often requiring more coordination to obtain groceries, as referenced in Table 3.

Additionally, the most commonly identified barrier was the cost of F/V. Participant 8 elaborated on increasing grocery prices affecting their ability to purchase F/V as follows:

“Because of the economy I don’t buy much for vegetables. There are times when I don’t but I should because I like the vegetables more than the meat. I really like to combine meat with vegetables, but at times I don’t have enough money to buy everything I would want to.”

Other participants also noted that they prioritized purchasing meat over F/V, indicating that meats are a cultural staple in a meal in comparison to F/V. However, Participant 8 also voiced concern over the prices of berries and grapes, stating, “the strawberries are really expensive. Because of that I can’t buy them, or the blueberries or even the grapes are expensive due to the season”.

Four participants noted challenges in obtaining their preferred F/V due to issues with transportation. This was connected to coordinating scheduled transportation with another individual to purchase groceries. Other participants noted challenges in finding their preferred cultural items, noting, “...I have to go to different stores but I do find them”, (Participant 2), as the familiar F/V were not all accessible in one place. However, the researchers noticed that the participants spoke more frequently about accessing F/V in general and less about accessing culturally appropriate or preferred F/V.

For the participants who noted that they lived alone (*n* = 3), there were increased barriers to accessing F/V. For example, Participant 1 showed strong cultural ties to serving F/V to their children and grandchildren; however, they noted that living alone (and cost, time, and transportation) has decreased their overall F/V consumption.

#### 3.3.3. Theme 3: Staple Items Are Used to Compliment Fruit and Vegetable Intake

Staple items, as part of the F/V kits, were curated to support the development of meals. The participants most frequently noted that they wanted rice (*n* = 15) and beans (*n* = 15), with a lower acceptance of *masa* flour, corn *tortillas*, and flour *tortillas*. The preference between canned and dried beans was explored, as there was a clear indication that participants did not want canned F/V. However, there was no clear indication from the eight participants that discussed a bean modality preference, as two participants indicated a preference for canned beans, four participants indicated a preference for dried beans, and two participants indicated that they did not have a preference. The participants advocated for the inclusion of onions, garlic, and cilantro for a more complete and robust staple package, with P6 expressing,

“...I feel like the onion, for me it’s important, the onion. It gives more flavor to the foods. I don’t know. We come from Latin America to here, and we lost a few items. Those flavors are different, and our diet changed, you know, until we adapt here to a different diet.”

#### 3.3.4. Theme 4: Variable Degrees of Health Literacy Are Present

The fourth theme emerged as the researchers noticed variable degrees of health literacy. More specifically, the knowledge of cooking and preparing F/V was high, whereas the portion sizes of F/V and the quantities recommended for good health were unclear. There was also an apparent resourcefulness and knowledge of the preservation of F/V that many individuals held.

When asking how certain F/V are prepared in the home, or whether the participant would require recipes to support F/V intake, all 15 participants noted ways in which they would cook items such as broccoli, spinach, and sweet potatoes (if they ate sweet potatoes). The majority of the participants noted that they would not need recipes for foods in Box A due to familiarity and high cooking literacy with all the items. However, Participant 9 was the only participant to note that vegetable consumption was not common for them while growing up. They showed an eagerness to learn how to prepare vegetables, and had good examples of how they used broccoli with multiple other vegetables, stating, “I boil the broccoli, and then I fry it with a little bit of jalapeño and that’s it. I do add mushrooms and onions [also]”. This participant did not think that they consumed enough F/V, even though they reported having a daily smoothie and reported consuming 4.7 servings of F/V daily, which is the highest average quantity of F/V consumed by participants in this study.

The participants were asked whether the boxes with ~50 servings would provide enough F/V for them to consume 8-10 servings of F/V per day, if added to what they are currently buying for their household. As a response, the participants commented, “that’s too many carrots in one day” (Participant 15), and “Yes, but 50 times! I think it will be breakfast, lunch, and dinner; it’s 5 days and 3 meals, that’s just 15 not 50” (Participant 6). Both patients were surprised by the quantities of F/V that would be appropriate to manage their chronic health condition via the DASH eating plan. Other participants asked for clarifications on serving sizes, asking, “For how many days? What’s a serving?” (Participant 2), indicating lower health literacy related to F/V portions.

During the interviews, the participants made note of how certain foods, especially fruits and other carbohydrate-containing foods, would affect their health. One participant had a strong sense of health advocacy around the foods that they “should not” be eating to manage their diabetes mellitus and had strong preferences for specific items that would be more helpful than others. Other participants lumped canned items with high sodium and/or sugar, and noted that because of their chronic health condition, they do not eat the canned items. Interestingly, these participants did not read the nutrition labels of the items or inquire about the sodium or sugar contents during the interview, but generalized that the canned items were not appropriate.

Many of the participants recognized that F/V are an important part of managing their chronic disease, and showed resourcefulness with regard to cooking/using the F/V or preserving them if they were not consumed. F/V were most commonly mentioned to be consumed via smoothies/shakes (*n* = 6), followed by ingredients in soup (*n* = 4). The participants emphasized that the food would not go to waste, and that they would find ways to eat it before it spoiled, as shown in Table 3.

## 4. Discussion

The objective of this research is to understand the cultural preferences, practices, and dietary habits of an immigrant, specifically in the Hispanic/*Latine* population. This understanding facilitates a greater adherence to the guidelines of the DASH eating plan to reduce markers of CVD, alleviate food insecurity, and enable the delivery of inclusive medical nutrition therapy to the Hispanic/*Latine* patient population. Learning from populations through a community-based participatory research approach is the key to providing and formulating appropriate dietary interventions. Moreover, as medically tailored meal kits increase in popularity, little attention is paid to cultural appropriateness and the accessibility of the intervention [33]. Given that the food insecurity rate in the im(migrant) Hispanic/*Latine* population in the Twin Cities metropolitan sampled in this study is 39.8% higher than the national average [34], it is important to consider the broader impacts that the F/V meal kits have on health, HTN management, and food access.

The findings from this study indicate that all participants preferred fresh fruits and vegetables over frozen and canned food options. The researchers recognize that further exploration and interviews are necessary to decipher the usage of the word “natural” and its connection to cultural preferences and practices surrounding fresh F/V in comparison to frozen or canned items. A study adding to the existing body of knowledge on the Healthy Immigrant Effect observed in Mexican immigrants who relocate to the U.S. underscores that these individuals maintain traditional dietary practices, including the consumption of whole grains, beans, and vegetables like tomatoes, onions, and peppers [35]. The participants in our study spoke of this change in availability and differing norms regarding canned versus fresh F/V in the U.S., challenging their ability to cook and experience foods in the capacity in which they are familiar. However, the results contradict those of other research that found the average Alternative Healthy Eating Index score, which is a measure of diet quality, increased for both men and women when living in the U.S. for more than 10 years in comparison to those that lived in the U.S. for less than 10 years [1]. This is of importance as we explore the connections of culturally appropriate foods that are accessible, affordable, and available within the U.S.

The data from our study show that most of the reported servings of F/V were consumed as vegetables, while fruits had a lower reported serving consumption. Consistent with the literature, our findings indicate that women reported a higher consumption of F/V than men on average, reiterating the belief that women are expected to eat more F/V, whereas men consume higher amounts of meat [36,37]. This could reflect as a higher score on the Healthy Eating Index, which parallels the aforementioned findings of Kershaw et al. [1]. Our research also shows an inverse relationship between the consumption of servings of F/V and education level (see Table 2), which is supported by the findings in a previous study assessing F/V intake in Hispanic/*Latine* populations [3]. However, our sample size did not allow for significance testing, limiting the formal evaluation of this relationship. Interestingly, a subset of these findings indicated that participants who experienced food insecurity had a higher intake of F/V in comparison to those who were food secure.

There was high acceptance of supplemental staple items to be provided with boxes for the increased participation and consumption of F/V in conjunction with the parameters of the DASH eating plan; thus, this brought us a step closer to developing a culturally appropriate meal kit. Our research shows that beans and rice were noted as the most essential staples for the composition of complete meals, with little interest in corn flour and pre-made flour *tortillas*, suggesting that participants may buy pre-prepared whole grain corn *tortillas* or may not use *masa* in their cultural dishes. Moreover, staple items tend to be cheaper in comparison to high-preference F/V, such as fresh guava, mango, tomatoes, and broccoli. As we found, this financial barrier can hinder the consumption of F/V and the adherence to dietary perimeters. This is echoed by the findings of Monsivais et al., with changes in dietary habits noted as being difficult when financial issues are present. Although the researchers described the DASH eating plan as being more expensive, their study’s results show that U.S.-residing Mexican American and Hispanic adults’ dietary patterns were more economically friendly in supporting the adherence to the dietary pattern. On a broader scale, this could translate to a wider acceptance of the DASH eating plan among Hispanic/*Latine* populations [38].

Although other barriers exist, the primary barrier is financial constraints, as it was noted as a challenge to accessing F/V by 60% of the participants. Financial barriers were reiterated by high rates of food insecurity in our sample. Sweeney et al. similarly identified cost as the main determinant factoring into food choices [33]. However, the complexities of acquisition and food choice is highlighted best by Dou et al., who cited education level, income, acculturation, and limited federal resource access as contributors to high rates of food insecurity among immigrant populations [2]. As increases in food prices continue to occur, as noted by the 9.9% increase in at-home food prices in 2022 alone [39], the authors recognize that the difficulties in affording F/V will be exacerbated for individuals with low incomes.

In addition to financial constrictions, time proved to be another barrier in the acquisition, preparation, and consumption of fresh F/V among our study population. Participant 7 expressed that medical appointments, travel time, family obligations, tiredness, and work obligations create inhibitions to consuming food on a regular schedule. Participant 6 explicitly stated the benefits of microwavable meals, noting the convenience of being able to prepare a meal in an average of 15–20 min, which was most desirable to accommodate their busy work schedule and lifestyle. While these expressions exist, most participants noted having a cooking time between 1–2 h, which coincides with previous findings from other research [33]. Cooking times and household numbers are unique to cultural traditions and social practices surrounding food. The average family size in which participants reported preparing meals for in our study was 3.6 people, which is consistent with the findings in other Hispanic/*Latine* populations [33]. While only one participant who lived alone expressed differences in the preparation of meals for a single person versus their family (Participant 11), further research is necessary to explore the relationships between the cultural norms, cooking practices, and social constructs of family involvement.

This work demonstrates the presence of important findings of variable degrees in personal health literacy, which is defined as an individual’s ability to make health-related decisions based on obtaining and interpreting the information provided to them [40]. The area in which confusion most often arose was portion sizes, where participants were challenged by what a serving of fruit or vegetables was and what qualified for each of these categories. Moreover, the participants recognized that canned items can be higher in sodium and/or sugar, but they did not inquire about the information provided on the Nutrition Facts panel to determine if the canned items shown in the box were low in sodium or in lite syrup. Wilson et al. investigated the associations between acculturation, quantified by the amount that Spanish was used as the primary language at home, and the use of the Nutrition Facts panel in Hispanic/*Latine* individuals. Their work found that the individuals who read the labels had significantly lower odds of poor diet quality compared to those that did not read the labels [41]. This emphasizes the importance of providing appropriate nutrition education to support individuals’ understanding of portion sizes and the Nutrition Facts panel to support health literacy.

The major strength of this study builds on the recent expansion in the use of medically tailored meal kits, which has shown to be effective in managing chronic conditions such as type 2 diabetes and heart failure via diet [15,16,17]. A scoping review regarding the assessment of different healthcare-based initiatives to improve access to F/V noted that there is promise regarding the effectiveness of medically tailored meal kits as an intervention on health outcomes; however, there is still a need for well-designed studies that support and validate the effectiveness of these interventions [42]. This is the first study to disseminate details regarding culturally appropriate nutrition interventions for individuals managing chronic diseases with limited resources using community-based participatory research methods, following the suggestions of Livingstone et al. with considerations for individuals from the Hispanic/*Latine* community to support the adaptation and tailoring of the intervention [14]. Moreover, this research helps to fill a gap in the literature regarding the acceptability of nutrition interventions such as medically tailored meal kits for individuals in Hispanic/*Latine* communities.

We recognize the limitations to this study, as our sample represents a small subset of the immigrant Hispanic/*Latine* population and is specific to a geographical location. Hence, we do not have an appropriate amount of information to make conclusions regarding diet patterns or the barriers to obtaining culturally appropriate foods for all Hispanic/*Latine* individuals. However, this methodology utilizing community-based participatory research methods could help to set frameworks to develop other health equity research, clinical interventions, and seed social changes for the mitigation of CVD in populations across the U.S. There was also potential for selection bias, as the participants volunteered to take part in the survey and the interviews, which may have increased bias towards those who were interested in F/V intake. Due to the convenience sampling method, not all of the participants in the study have been diagnosed with CVD, but instead, have been diagnosed with a chronic disease; therefore, we could not speak to the experiences and desires of those with CVD. Additionally, the participants did not receive specific education regarding the cup sizes of F/V; thus, they may not have been properly estimated when completing the survey. However, the qualitative and quantitative data assessed in this study provide a voice for Hispanic/*Latine* individuals facing high rates of chronic diseases and food insecurity. More research and action in nutrition-related policy are needed to address the barriers that immigrant Hispanic/*Latine* individuals may face in accessing culturally appropriate F/V. More attention should be paid to the voices of communities to develop culturally appropriate nutrition interventions that meet the recommendations to effectively manage chronic diseases, for example, HTN through a DASH eating plan. This tailored intervention has strong potential for future research, as we highlighted the variables that could be used in the treatment of CVD for Hispanic/*Latine* populations. There is possibility for its application in future work to quantify changes in the markers of CVD in a clinical setting using medically tailored food kits as these dietary interventions increase in popularity and show continued effectiveness for chronic disease management.

## Figures and Tables

**Figure 1 nutrients-15-03600-f001:**
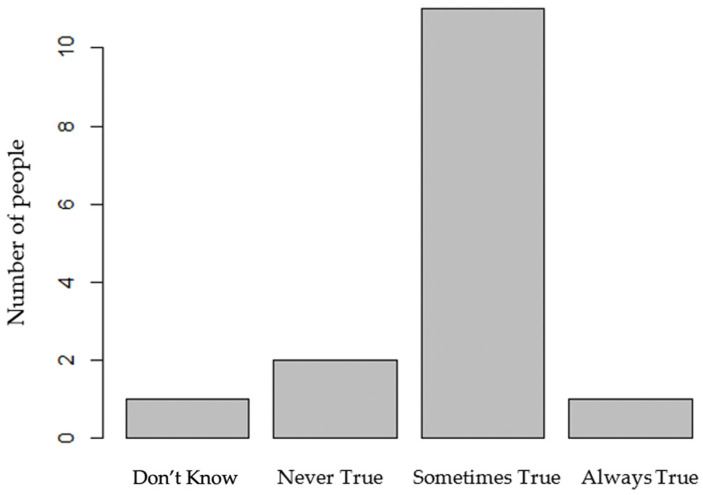
Food insecurity summary showing details of those that could not afford balanced meals.

**Figure 2 nutrients-15-03600-f002:**
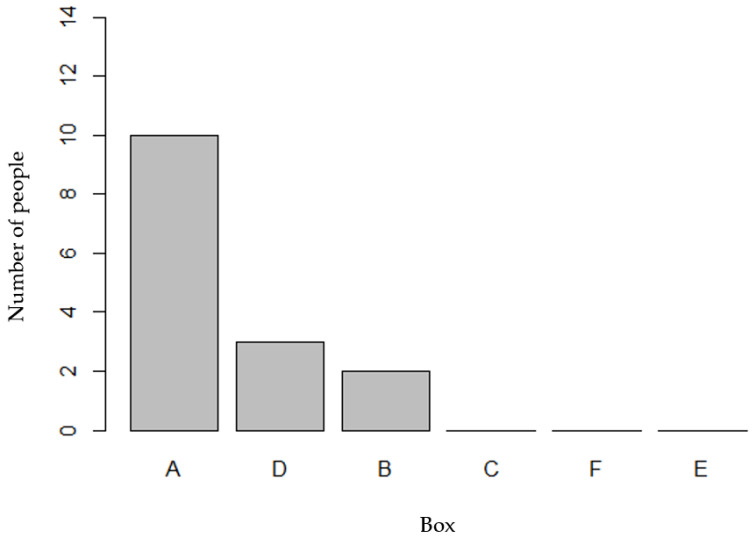
Number of individuals who chose each medically tailored food box as their first preference.

**Figure 3 nutrients-15-03600-f003:**
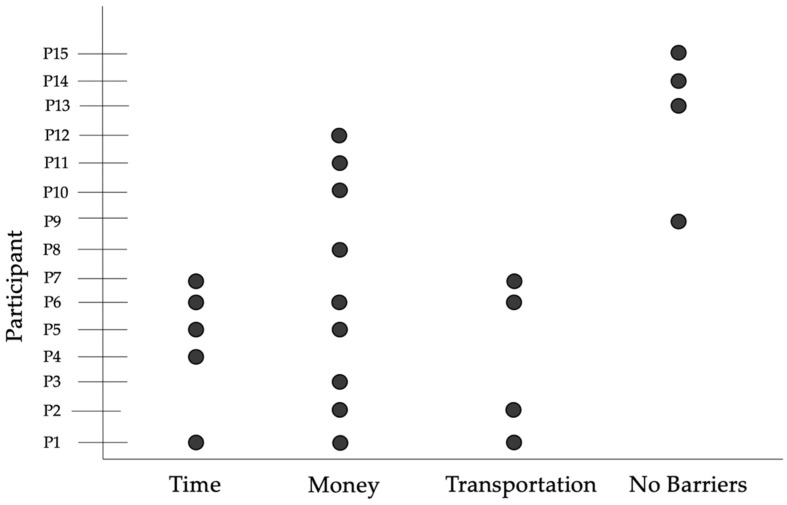
Beeswarm plot showing number of participants indicating that time, money, and/or transportation were barriers to accessing fruits and vegetables for their household. Each line represents an individual participant, and a dot represents a positive indication of the specified barrier being present for that individual.

**Table 1 nutrients-15-03600-t001:** List of fruit and vegetable items represented in each box, including modality and servings provided.

Box Label (Total Servings)	Fruits, Modality Provided (Servings)	Vegetables, Modality Provided (Servings)
Box A (47)	1 mango, fresh (2) 1 avocado, fresh (2) 3 bananas, fresh (3) 1 pound of grapes, fresh (8) 3 large apples, fresh (6)	3 tomatoes, fresh (3) 2 cucumbers, fresh (4) 2 chayote squash, fresh (2) Broccoli, fresh (4) Spinach, fresh (3) 1 head of lettuce, fresh (2) 8-10 carrots, fresh (8)
Box B (56)	Mangoes, frozen (6) Strawberries, frozen (4) 1 avocado, fresh (2) 3 bananas, fresh (3) 1 small watermelon, fresh (10)	3 tomatoes, fresh (3) 2 cucumbers, fresh (4) 8-10 carrots, fresh (8) Broccoli, frozen (8) Asparagus, frozen (8)
Box C (52)	Mangoes, frozen (6) 1 avocado, fresh (2) 3 bananas, fresh (3) 1 pound of grapes, fresh (8) 6 single servings of applesauce, canned (6)	Diced tomatoes, canned (3) 2 cucumbers, fresh (4) 8-10 carrots, fresh (8) Broccoli, frozen (8) Corn, canned (4)
Box D (50)	1 mango, fresh (2) 1 avocado, fresh (2) 3 oranges, fresh (3) 1 cantaloupe, fresh (8) 5 peaches, fresh (5)	3 tomatoes, fresh (3) 2 bell peppers (red/green), fresh (4) 1 butternut squash, fresh (8) Broccoli, fresh (4) Spinach, fresh (3) Kale, fresh (2) 6-8 carrots, fresh (6)
Box E (49)	Mangoes, frozen (6) Peaches, frozen (4) 1 avocado, fresh (2) 3 oranges, fresh (3) Strawberries, frozen (4)	3 tomatoes, fresh (3) 2 bell peppers (red/green), fresh (4) 4 sweet potatoes (4) Broccoli, frozen (8) Spinach, frozen (3) Kale, fresh (2) 6-8 carrots, fresh (6)
Box F (49)	Mangoes, frozen (6) Papaya, fresh (4) 1 avocado, fresh (2) 3 oranges, fresh (3) Pineapple, canned (2) Mixed fruit (peaches, mango, pineapple, and strawberries), frozen (2)	Diced tomatoes, canned (3) 2 bell peppers (red/green), fresh (4) 2 sweet potatoes, fresh (4) Broccoli, frozen (8) Spinach, frozen (3) Kale, fresh (2) 2 pumpkins, canned (6)

**Table 2 nutrients-15-03600-t002:** Average combined number of servings of fruits and vegetables, as assessed using the 10-question Dietary Screener Questionnaire, reported by participant demographics (*n* = 15).

Variable (*n* = 15)	*n* per Category (Percent Sample)	Mean Servings of Total F/V per Day (SD)
Sex		
Female	12 (80)	2.7 (1.4)
Male	3 (20)	2.2 (1.4)
Education		
Some High School	6 (40.1)	3.3 (1.6)
High School Graduate	2 (13.3)	1.6 (0.3)
Some College, No Degree	2 (13.3)	1.3 (0.4)
Bachelor’s Degree	2 (13.3)	2.7 (1.4)
Professional Degree	1 (6.7)	1.8 (0.0)
Not Reported	2 (13.3)	3.3 (0.6)
Employment		
Housekeeping	2 (13.3)	2.5 (0.7)
Looking for Work	4 (26.7)	2.1 (1.3)
Working Now	6 (40)	2.6 (1.4)
Retired	1 (6.7)	3.8 (0.0)
Temporarily Laid Off	1 (6.7)	5.2 (0.0)
Other	1 (6.7)	1.8 (0.0)
Chronic Disease Diagnosis *		
Hypertension	10 (66.7)	2.3 (1.2)
Dyslipidemia	5 (33.3)	2.5 (1.2)
Pre-Diabetes	3 (20.0)	2.0 (0.9)
Type 2 Diabetes Mellitus	6 (40.0)	3.7 (1.2)
>1 Chronic Disease	7(46.7)	2.7 (1.2)

* Participants were included in each group if they indicated a diagnosis of the chronic disease. Participants indicating >1 chronic disease show overlap across multiple categories.

**Table 3 nutrients-15-03600-t003:** Thematic map summarizing themes and subthemes related to preferences for fresh F/V, barriers to accessing F/V, staple items to support F/V intake, and degrees of health literacy from 15 interviews with Hispanic/*Latine* individuals managing chronic diseases.

Theme	Subtheme	Quotation
Fresh foods and vegetables are top food preference, followed by frozen fruits and vegetables.	Frozen food is recognized as “natural” and can be preserved longer than fresh F/V. Canned foods are reserved as a last resort and can be perceived to have chemicals and/or be high in sodium and sugar. Certain cultural canned foods like corn and beans are accepted.	Participant (P)7: “...the best option is the most natural because of the chemicals that the canned goods have.”
		P5: “Personally, when I just arrived to the United States, we didn’t consume much of the canned products. Each time someone would go to the store close to home, there would be fruits, *elote*, or fresh tomatoes; it was natural not canned. It was hard for me to cook with the canned food because it has another type of flavor.”
	Individuals have a like or dislike for fresh fruits and vegetables based on personal food preferences and/or cultural upbringing.	P14: “...For example, I would like to exchange [the kale] because why bring it if I’m not going to eat it, or maybe end up throwing it in the trash.”
		P2: “I know the [chayote] in the corner, it might not have a lot of nutrition, but you can’t really find it here. It’s appropriate for my culture. I know everything, I use everything, mostly daily.”
	F/V commonly used in shakes, smoothies, or soups.	P3: “I like to make the shakes with spinach or fruits.”
		P7: “...The carrot, and the spinach and even kale or oranges– I have made shakes or juices, and I put cucumbers [too].”
Barriers to access F/V in order include time, money, transportation, and living alone.	Time: Participants spend 20 min and up to 1 h preparing food.	P7: “I have medical appointments. Sometimes I come back home tired and I lay down. Sometimes I eat something. There have been occasions where I rest, and then I go pick my daughter up. I leave early to work, really tired, and I eat at 9 pm. It depends on my routine. When I have appointments, it’s hard with time and the location. The appointments are lengthy. I don’t have a specific time to eat.”
		P6: “I don’t have too much time, I use the microwave as much as possible, I try to do it 15, 20 min and that’s it or less, I’m always in a rush.”
	Money: Groceries are expensive, particularly F/V.	P10: “Since I’m looking for a job, I try to keep calm, but I don’t think too much. I won’t die of hunger, there’s a lot of water, with water one can maintain.”
		P5: “If there was a better price, I could buy them more because sometimes income is a little bit limited. Sometimes at home there isn’t much to eat but vegetables and a little bit of meat or a small piece of *chorizo* or an egg and [the kids] whine. So, I try to buy a piece of [meat] and either rice or beans and a little bit of lettuce salad with slices of tomato, and onions, which would be the most common. I try to make all of that but I can’t buy all the fruit or the vegetables.”
		P2: “What would help me eat more fruits and vegetables? Money.”
	Transportation: A lack of access to transportation is a barrier to obtaining F/V.	P1: “My children are the ones that take me and bring me to doctor appointments. They work, and I work on the weekdays.”
	Living alone is a potential risk factor of cooking less.	P11: “I don’t make a lot of elaborate food, only when my kids come over, I make them something special. For me I try to make my foods practical, not really elaborate, but simple so that it’s good.”
		P1: “I live alone and I battle a lot to bring the things.”
Staple food items identified as central.	Onions, garlic, rice, beans, and cilantro are important staples.	P12: “Onions—because it isn’t typical to cook the other foods in the boxes without onions. You need onions.”
		P6: “We use onions, that’s missing there”
		P4: “The rice—I use it daily; my kids eat rice every day. Beans—well every week, or all the items every week.”
	Canned items are not a top preference, except for beans, which have a mixed preference in terms of dried versus canned beans.	P5: “Because we consume *tortillas*, for the beans we consume them as canned and also cook them. The rice is also a base food because you accompany it with meats, sometimes with beans or eggs. So as much as the beans, for us, it’s a compliment that doesn’t miss, it’s to eat every day.”
		P6: “I don’t cook much of the beans, if I eat it, I eat it from the cans.”
		P11: “Yes, I like the beans in the bag, so I can cook them here at home.”
Variable degrees of health literacy are present.	Cooking literacy among the population is high.	P11: “I like to cook a lot, and I know how to do a lot of things with the vegetables.”
	Awareness of how to manage chronic diseases and how certain foods, including cultural dishes or sugar, may impact disease management.	P9: “...I buy them usually—vegetables and the fruits for the kids, and the vegetables so I can eat them for my diabetes. Now I’m doing that because before I didn’t use to eat them.”
		P10: “Look *tortillas*, rice and beans would be good, because there is a mix, a lot of *tortillas* I haven’t eaten since I joined [the clinic]. What I thought was good for me was not, now I eat a little bit of *tortillas*, rice and beans are good.”
		P10: “I eat green beans as well, and mushrooms as well. That’s why my levels went down, I used to be very bad, and now not as bad.”
		P5: “*Piloncillo* [Panela, whole cane sugar] but since I’m diabetic and I can’t eat it I prepare it in the oven, or boiled, and I put some sugar or milk, Nestle, and a bit of cheese and we eat it as if it was bread.”
	Individuals are resourceful.	P12: “For me the food would not get spoiled. We would eat it, you know. I know how important it is. And I use so many of these to blend or juice. So, nothing would get spoiled—it has to get into my system.”
		P7: “Sometimes I make juices in the blender, using sweet potatoes, and honey they’re good, better than the carrot ones. If I don’t have, and I only have carrot, then I make it with carrots, if I have celery then I only use celery.”
		P13: “...Everything that’s in there I would eat because I won’t waste food.”

## Data Availability

The data presented in this study are available upon request from the corresponding authors. The data are not publicly available due to sensitive information regarding the documentation status of immigrants.

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
