# Peer review of "Using Community-Based Participatory Research Methods to Inform the Development of Medically Tailored Food Kits for Hispanic/Latine Adults with Hypertension: A Qualitative Study"

_nutrients, 2023, doi:10.3390/nu15163600_

Round 1

Reviewer 1 Report

Dear Authors I report my considerations regarding your paper:

Abstract: 

-line 2 you cite the CVD for the first time put in brackets (cardiovascular disease). Remove cardiovascular disease in the next abstract sentences putting CVD.

2.1 Study design 

2.2 Partecipants and recruitment 

"Out of the 60 partecipants 23 consented to participate the study"

Are you saying that you enrolled 23 patients in total and only 15 completed the study? 

The biggest flaw the study is related to the number of patients. As a reviewer is hard for me find a possible scientific application regarding your work. The title is:

"Using community-based participatory research methods for the development of medically-tailored food kits for Hispanic/Latine patients with hypertension". 

Considering your title, how do you understand that your work can be useful to mitigate the CVD risk for Hispanic/Latine populations?

I explain myself better, in your results I can't see any possible useful informations that could be used to reach your final goal. At the end you reported the interviews obtained from your 15 patients after the development of a personalized food kit. However, how do you know that this type of  diet will help patients with their diseases? Considering that you obtained these data from 15 people and you didn't present any type of clinical data? 

3. Results

Partecipant demographics

How do you think that these result could represent the hyspanic/latin population? Considering that you had only 3 males in the study? 

There is a big flaw and also bias from representing the entire hyspanic/latin population with 3 males.

Table 3 

Regarding this table I don't understand the sense to report the quotations of the patients, how can help their statements regarding the final aim of the paper. You could present their cultural dietary practises in other ways in a less personal and more scientific way. 

Figure 3 

I appreciate the time, money and transportation considerations. These are three important barriers to obtain high quality food, however also in this case we are talking about fifteen people. From statistically point of you is not revelant, especially if you think that all your datas come from  questionnaires. 

Discussion 

The main problem of your work is the size of the sample and you also know very well: 

"However, our samples size did not allow for significance testing result"

So, my question is why are you presenting to us something that is not significant? 

The topic that you faced is very important but the method that you used and the way of presenting your data unfortunately is opinable.

You conclude the paper with this statment: 

"Food kits increase in popularity and show continued effectiveness for chronic disease management" 

Ok, but my question is how do you know that your medically tailored food kits are effective? Did you have some datas regarding this statement? How can you suggest to others your personal plan without other scientific measurements? 

"This study has a strong application in a clinical setting" 

How can you prove it this statements with your datas? Where are these strong applications? 

Unfortunately you reported only some sentences from the patients pointing out some problems that this type of community could face it, I say could because we are speaking about fifteen people. 

Reviewer 2 Report

Thank you very much for giving me the opportunity to review this paper. Congratulations to the authors for the work done, here I attached with some revisions and proposals for improvement. 

Using community-based participatory research methods for the development of medically-tailored food kits for Hispanic/Latine patients with hypertension

NUTRIENTS JOURNAL

COMMENTS – Comments and Suggestions for Authors

STRUCTURE

-        The manuscript is not correctly structured.

TITLE AND ABSTRACT

-        The Abstract is properly structured. The title is concise, specific and relevant. Type of intervention is indicated (Dietary Approaches to Stop Hypertension).

-        The title does include the study population. However, it would be interesting to know the age range at which the questionnaire is aimed.

-        The title or abstract should give information on what kind of study it is about. It is also advisable to add the population where the study has been carried out.

-        The Abstract is properly structured, but authors can include in each section their sub-section in bold, e.g. Background: " The Dietary Approaches to Stop Hypertension (DASH) eating plan is the most effective dietary intervention for CVD, but ...". In the same vein, materials and method, results and conclusions would be indicated.

-        CVD: Describe each abbreviation the first time it is mentioned in the text.

INTRODUCTION

-        The introduction shows the importance of the study. And the references used are recent, but is recommended to emphasise the prevalence of obesity and associated morbidity and mortality. to find out more about chronic diseases, what factors trigger them, for example obesity. Excess weight can compromise the health of citizens with chronic non-communicable diseases.

o   World Health Organization. U.S. Department of Health and Human Services. National Institute on Aging. National Institutes of Health;Global Health and Aging. 2011 :1–32. Available online: https://www.nia.nih.gov/sites/default/files/2017-06/global_health_aging.pdf

o   Micha R, Peñalvo JL, Cudhea F, Imamura F, Rehm CD, Mozaffarian D. Association Between Dietary Factors and Mortality From Heart Disease, Stroke, and Type 2 Diabetes in the United States. JAMA. 2017 Mar 7;317(9):912-924. doi: 10.1001/jama.2017.0947. PMID: 28267855; PMCID: PMC5852674.

-        It is recommended that the hypothesis under study be added at the end of this section. The hypothesis of the work carried out should be explicitly stated.

-        A study examining the link between length of residence in the U.S. and CVD risk for Hispanic/Latine individuals also found gender differences: women who migrated to the U.S. within 10 years had lower CVD risk, whereas men exhibit the lowest CVD risk after living in the U.S. for more than 10 years [1].” It is recommended to investigate these gender differences, to what can they be due?

-        They state that the questionnaire is aimed at older adults. However, it would be interesting to know the age range to which the questionnaire is addressed. For example: “PROMIS® (Patient-Reported Outcomes Measurement Information System) is a set of person-centered measures that evaluates and monitors physical, mental, and social health in adults and children. It can be used with the general population and with individuals living with chronic conditions.

o   PROMIS Scale v1.2 - Global Health. 2023. Available online: https://www.healthmeasures.net/index.php?option=com_instruments&view=measure&id=778&Itemid=992

-        The PhenX questionnaire is included in this study, however, there is no mention of its "wide acceptance, as the use of PhenX measures will promote comparisons between studies to increase statistical power to identify and replicate variants associated with complex diseases and gene-gene and gene-environment interactions.

o   Hamilton CM, Strader LC, Pratt JG, Maiese D, Hendershot T, Kwok RK, Hammond JA, Huggins W, Jackman D, Pan H, Nettles DS, Beaty TH, Farrer LA, Kraft P, Marazita ML, Ordovas JM, Pato CN, Spitz MR, Wagener D, Williams M, Junkins HA, Harlan WR, Ramos EM, Haines J. The PhenX Toolkit: get the most from your measures. Am J Epidemiol. 2011 Aug 1;174(3):253-60. doi: 10.1093/aje/kwr193. Epub 2011 Jul 11. PMID: 21749974; PMCID: PMC3141081.

-        In addition, it would be relevant to justify why the questionnaire is aimed at this type of population and not others.

-        In general, the introduction is quite comprehensive, and shows a good overview of the current state of the subject.

MATERIAL AND METHODS

-        This section includes different relevant data of the study without differentiating it by sections, it is advisable to separate it with the following sections: design, inclusion criteria, statistical analysis, etc.

2.1. Study design

-        it is recommended that the "study design" section be separated into two subsections. The first with detailed information on the participants, including inclusion and exclusion criteria. The other section with the appropriate information on the survey.

-        Table 1. It is not necessary to put servings in brackets again since the first row already mentions what it means in brackets.

-        Low cardiovascular risk depends on all of the following criteria: total serum cholesterol <200 mg/dL and no cholesterol-lowering medications; systolic/diastolic blood pressure <120/<80 mm Hg and no antihypertensive medications; BMI <25.0 kg/m 2 ; fasting plasma glucose <100 mg/dL and no history of diabetes mellitus; no current smoking; and no major electrocardiogram abnormalities. Were these cutoff points (according to the American Heart Association criteria for ideal cardiovascular health) taken into account when making the exclusion and inclusion criteria?

o   Kershaw KN, Giacinto RE, Gonzalez F, Isasi CR, Salgado H, Stamler J, Talavera GA, Tarraf W, Van Horn L, Wu D, Daviglus ML. Relationships of nativity and length of residence in the U.S. with favorable cardiovascular health among Hispanics/Latinos: The Hispanic Community Health Study/Study of Latinos (HCHS/SOL). Prev Med. 2016 Aug;89:84-89. doi: 10.1016/j.ypmed.2016.05.013. Epub 2016 May 16. PMID: 27196144; PMCID: PMC4969108.

-        It is recommended that the first table be related to the sociodemographic characteristics of the sample included.

-        Another final subsection could also be introduced, so that the other implemented questionnaires could be added. With a brief description of each of them, and their respective references.

o   DSQ allows the following to be obtained with respect to the foods requested: The questionnaire includes a list of common foods and beverages, categorized into different groups (e.g., dairy, meats, fruits, vegetables, grains, etc.). Participants must indicate how many times they consumed each food or beverage during a given period of time (usually in the last 30 days). Why are these results not shown in a table in the RESULTS section?

-        Why are these products included in the F/V group?: “Staple items such as rice, masa flour, beans, tortillas, onions, cilantro” etc.,

-        The survey was completed by 50 patients or providers that self-identified as Hispanic/Latine from a community health clinic.” A sample of 50 people is indicated here, however in the abstract 15 are listed. Check that this is correct.

-        Add the complete name of the questionnaire - reference 19-:

o   National Institute of Health, National Cancer Institute. Dietary Screener Questionnaire (DSQ) in the NHANES 2009-10: Dietary Factors, Food Items Asked, and Testing Status for DSQ. Division of Cancer Control and Population Science. Available online: https://epi.grants.cancer.gov/nhanes/dietscreen/evaluation.html#pub (accessed on October 5, 2022).

2.3. Data collection

-        This method is particularly effective to provide formative interventions and cross-cultural understanding [20]”. Add this information in the introduction section. The methodology section includes information related to the instruments used and the methodology to be followed.

-        Since the 7-step framework method is followed, the authors are advised to describe specifically what each of the steps consisted of, what was carried out, and in what time period each step took place.

-        The research design and methodological approach used in this qualitative study are described.

-        Data collection: Describes in detail the data collection techniques employed. Explains how they were carried out and how the data were recorded.

-        Data analysis: Explains how the qualitative data collected were analyzed. Mention the coding, categorization and thematic pattern search approaches used. However, the statistical analysis is not shown: was the mean, standard deviation, confidence interval calculated? This is very important, especially if you want to know if the sample population has statistically significant differences with respect to some variables included in this study (e.g. sex, age...). What was the p value used then?

RESULTS

-        Table 2. Homogenize the results. If you write "percent sample" then why do you put it in symbol (%)? On the other hand, if the numbers have a decimal place, it should always be like this. It is recommended to put n=15 in the first row of the table.

-        Figure 1. frequency (Y axis) refers to the number of people?

-        Table 3. When a table is split into two sheets, the sections must be put back in the header. Applicable to the rest of the manuscript. Include a margin of separation between the subtopic and the participant's quote. In this way the reader will differentiate between each subtopic and what the participant is referring to. Below the table include the meaning of the abbreviations used. Applicable for the rest of the document.

-        Figure 3. In figure 3 it is not clear whether the point is each participant.

DISCUSSION

-        Including the language limitation, did this factor hinder the study?

-        For future research, it is recommended to add future interventions and/or proposals for improvement, especially in relation to the difficulty/barrier with respect to money and time. What feasible facilities can be offered?

CONCLUSION

-        Conclusion (optional): has been included. Researchers have made a brief conclusion in this section.

REFERENCES

-        References are numbered in order of appearance in the text. They are correctly placed in square brackets [ ] and placed before punctuation.

-        The reference section does not follow the recommended style. The authors' names are not well written. Below is an example of how it should be:

o   Gnagnarella, P., Dragà, D., Baggi, F., Simoncini, M. C., Sabbatini, A., Mazzocco, K., Bassi, F. D., Pravettoni, G., & Maisonneuve, P. (2016). Promoting weight loss through diet and exercise in overweight or obese breast cancer survivors (InForma): study protocol for a randomized controlled trial. Trials, 17(1). https://doi.org/10.1186/S13063-016-1487-X

-        A bibliographic manager should be used so that the citations are well inserted, homogeneous and in accordance with the standards recommended by the journal. It is important that the year appears in bold and the journal in italics. Check the references because many of them are wrong (names, rules of the references, etc.).

In terms of language some terms such as *tortillas have to be modified.

Round 2

Reviewer 1 Report

Dear authors, 

first of all thank you for your kind replies. I would like to discuss better regarding your answer and my doubts.

I ask you an additional information regarding the material and method: 

"This process included a conceptualization component, which was carried out via a literature informed survey completed by patients or providers self-identifying as Hispanic/Latine from a community health clinic to determine F/V preferences. This survey informed the formative intervention for 1-on-1 interviews to learn about the perceptions of the DASHappropriate food kit design. Six F/V boxes consisting of culturally-appropriate F/V indicated as higher preference were developed. High preference items were denoted as F/V that over 75% of the survey population indicated as preferred in the survey".

Can I ask you how many people were used in this part of the study? I don't understand if you reported this information elsewhere. Thank you, if this information is missing please add it also in the previous section cause is not clear. In this specific point my question was over 75%, okay but 75% of which number? 

Point 2

The purpose of this study is to discuss the qualitative components related to the food kits and barriers to obtain fruits and vegetables in their lives. If you require more clarification, please feel free to make specific suggestions.

Totally understandable, however this is not a problem related only to the hispanic culture. I think that your component are also  related to the entire nation, in fact according to the USDA in United States the consumption of vegetable and fruit is under the 2020-2025 dietary reccomandations guidelines. So, how can you attribute this problem only to the hispanic population? In this way seems that only the hispanic culture suffer regarding this type of problematic. What i am saying is that from this point of view I don't see the novelty of the paper, cause a lot of different ethnicities are facing these issues.

This point is strictly connected to the point 7 where you reported as a crucial barrier to have a good F/V ratio.

Theme 2: Time, Money, and Transportation

Also in this case Time, Money and Transportation are three variables which affect the entire American Population. A lot of people which are not Hispanic or latin can't have the right F/V for this three type of barriers even if I find very interesting and current this consideration. 

Point 5

Because we only had participants from the Hispanic/Latine population, we have tailored this intervention to the patients. We are not claiming this represents all of this population.

You are not claiming to represents all of this population but at the same time you assess that:

the primary outcome is "to understand barriers to obtaining culturally-appropriate F/V and identify F/V preferences that align with the DASH eating plan for the development of a medically-tailored kit for immigrant Hispanic/Latine individuals with HTN".

So the primary purpose is to improve and develop a better DASH kit for Hispanic/Latine individuals. Is it correct? So you are claiming that with your data you are not representing all the Hispanic/Latin population. That is one of the reason why I don't understand which is the final aim of the paper and I don't think that the method sustain the final aim. I think there is a contradiction between the methods applied and the final aim of the work and this because you are trying to suggest a new model development of a medically-tailored kit for immigrant Hispanic/Latine individuals with HTN with a preliminary study.

Conclusions

"This formative intervention has strong potential for future research. There is possibility for application in future work to quantify changes in markers of CVD in a clinical setting using medically-tailored food kits as these dietary interventions".

If you consider this sentence it seems that your paper represent the starting point to apply immediately in the next studies your new dietary approach, however I consider this study as a preliminary study where you considered the possibile variables which could be involved in the CVD treatment for Hispanic Population with hypertension thanks to the use of personalized DASH dedicated to this specific ethnicity. 

Reviewer 2 Report

Thanks to the authors for their work and modifications. Some aspects need to be improved after this second revision. New comments after this second review are shown in red. 
